*Report*

# Microscopy Nodes: versatile 3D microscopy visualization with Blender

Aafke Gros [1✉], Chandni Bhickta[1], Granita Lokaj[1], Brady Johnston [2], Yannick Schwab [1], Simone Köhler[1] & Niccolò Banterle [1✉]

## Abstract

**Effective visualization of 3D microscopy data is essential for communicating biological results. While scientific 3D rendering software is specifically designed for this purpose, it often lacks the flexibility found in non-scientific software like Blender, which is a free and open-source 3D graphics platform. However, loading microscopy data in Blender is not trivial. To bridge this gap, we introduce Microscopy Nodes, an extension for Blender that enables the seamless integration of large microscopy data. Microscopy Nodes provides efficient loading and visualization of up to 5D microscopy data from Tif and OME-Zarr files. Microscopy Nodes supports various visualization modes including volumetric, isosurface, and label-mask representations, and offers additional tools for slicing, annotation, and dynamic adjustments. By leveraging Blender's advanced rendering capabilities, users can create high-quality visualizations that accommodate both light and electron microscopy. Microscopy Nodes makes powerful, clear data visualization available to all researchers, regardless of their computational experience, and is available through the Blender extensions platform with comprehensive tutorials.**

**Keywords** Data Visualization; 3D Data; Fluorescence Microscopy; Electron Microscopy; Blender
**Subject Categories** Computational Biology; Methods & Resources

## Introduction

Biology is three-dimensional and dynamic, but most representations are limited to two dimensions. The informative and correct mapping from 3D space to 2D visualization, 'rendering', is therefore essential for communicating biological 3D data. Multiple scientific software solutions exist for rendering data in a 3D environment, both proprietary, such as Imaris (Imaris), Arivis (Arivis) and Amira (Amira), and open-source alternatives such as BigVolumeViewer (Pietzsch et al, 2015; bigdataviewer), Agave (Toloudis and Contributors, 2025) and Napari (Chiu et al, 2022).

However, these scientific tools are often limited to workflows for specific imaging modalities, where editing the rendering pipeline or 3D scene for custom pipelines remains challenging. By contrast, non-scientific 3D rendering software has had a large user base and need for diverse features for a long time, leading to comprehensive and mature software packages in both the proprietary and open-source space.

One of these non-scientific solutions for 3D rendering is the powerful software Blender (Blender), which is committed to staying free and open-source long-term. As a generic graphics toolkit, Blender does not specifically support easy loading of microscopy data, nor can it readily handle the large data size of microscopy volumes. Although Blender has been used previously by microscopists, it was only accessible to users with extensive knowledge of both Python and the Blender user interface (for example, Hennies et al, 2023). Here, we present Microscopy Nodes, a bridge between the microscopist and advanced data representation in Blender. Microscopy Nodes seamlessly integrates microscopy data into an ecosystem, providing powerful tools for beautiful presentation-quality visualizations, enabling more effective communication of 3D biological data.

## Results and discussion

Using Microscopy Nodes, we can load a large variety of microscopy modalities and target a diverse set of output visualizations. Here, we showcase a selection of different target visualizations that can be achieved with Microscopy Nodes and Blender. The input modalities tested include real-time imaging (Fig. 1A, https://uk1s3.embassy.ebi.ac.uk/idr/zarr/v0.4/idr0052A/5514375.zarr), larger datasets, such as a 49 GB expansion microscopy stack (Fig. 1B, https://uk1s3.embassy.ebi.ac.uk/idr/share/microscopynodes/20250730/RPE1_4x.zarr, generated for this showcase), super-resolution microscopy (Fig. 1C, https://uk1s3.embassy.ebi.ac.uk/idr/share/microscopynodes/20250730/UExSTED.zarr) and a 14.5 GB electron microscopy dataset (Fig. 1D, EMPIAR-11399). These datasets are hosted publicly in archives such as the IDR (Williams et al, 2017), Bioimage archive (Hartley et al, 2022), and EMPIAR (Iudin et al, 2023). Microscopy Nodes can load local Tif or OME-Zarr files, or, when available, OME-Zarr datasets from their corresponding address (URI) in repositories. This feature enables interested users to make use of Microscopy Nodes even without their own dataset.

[1]European Molecular Biology Laboratory, Cell Biology and Biophysics Unit, Heidelberg, Germany. [2]School of Molecular Sciences, The University of Western Australia, Perth, WA, Australia. ✉E-mail: aafke.gros@embl.de; niccolo.banterle@embl.de

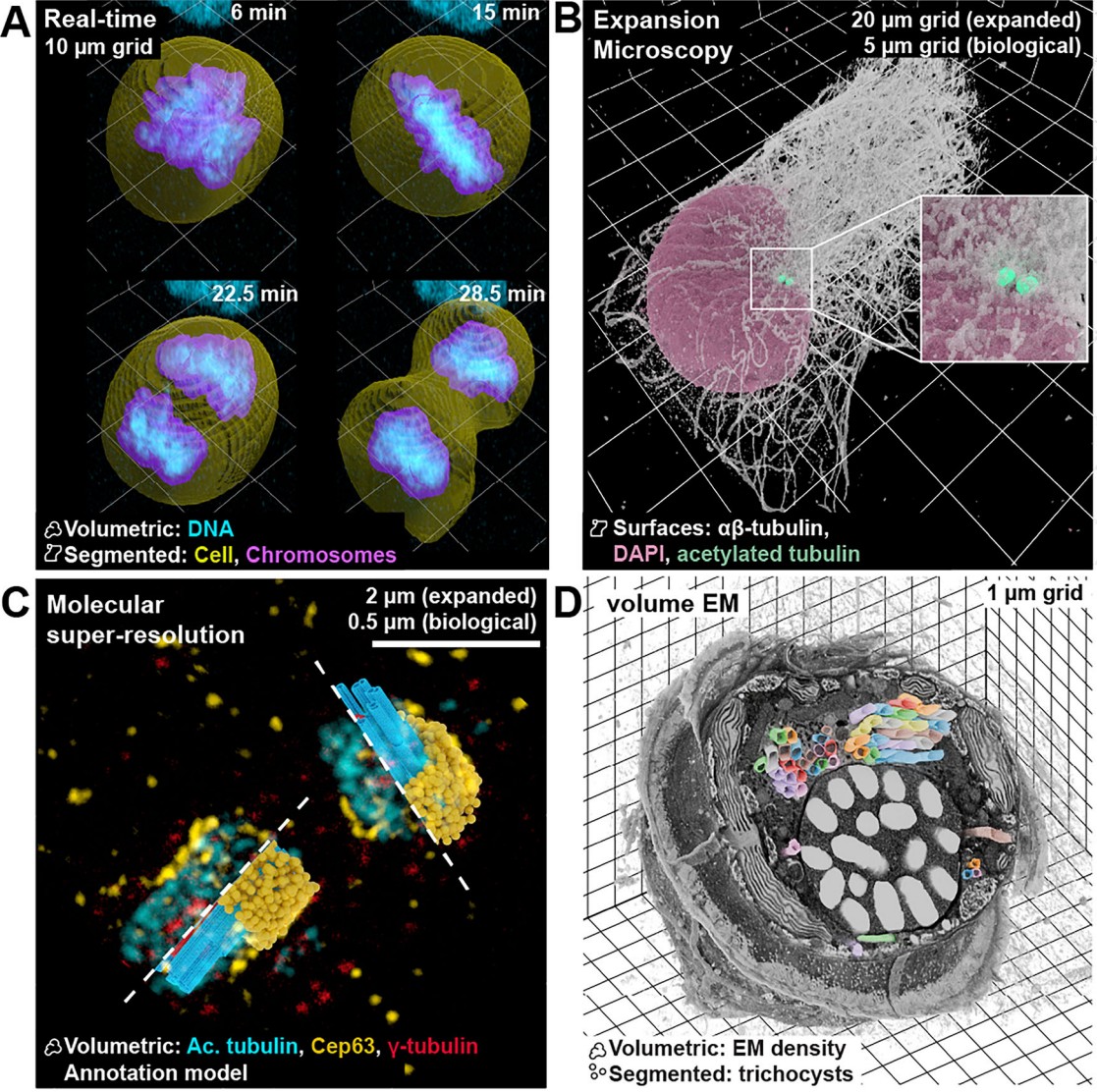

**Figure 1. Microscopy Nodes supports diverse data modalities and output formats.**

(A) A time series of a mitotic cell with segmentations. The render shows the dynamics of chromosomes (DNA) during mitosis by combining mesh and volumetric rendering modes of data from Walther et al (Walther et al, 2018). (B) An isosurface render of a RPE1 cell shows its intricate cytoskeleton. The expansion microscopy image of the cell is shown as isosurfaces with the αβ-tubulin slightly transparent to highlight the centrioles (acetylated tubulin). The versatility to independently adapt separate channels allows for better communication of the data. (C) Blender allows for the integration of models and data into the same space. U-Ex-STED imaging of centriolar components, with an integrated procedurally generated model of the centriole, shown for half of the centrioles each. (D) Blender allows for contextual EM visualization. A dinoflagellate cell imaged with FIB-SEM and its segmented trichocysts (Mocaer et al, 2023). With Microscopy Nodes and Blender, we can reliably visualize the ultrastructure of cells and show EM data where electron-sparse regions are transparent, with segmentations in the same view. Here, the segmentations are cropped to a separate region from the EM density, to show them protruding from the volume.

When loading data with Microscopy Nodes, multiple representations are available, and which representation is appropriate depends on the type of data. For example, Fig. 1A and Movie EV1 show a real-time imaging dataset with fluorescent channels and segmentations of a cell and its chromosomes. For this dataset, we visualize one of the fluorescent channels (DNA) as a volumetric and emissive render, meaning that each voxel emits light with an intensity proportional to its measured fluorescence value in the dataset. This way of handling fluorescent data matches the

acquisition technique by representing the light-emissive fluorescent probe as a light-emissive signal in the 3D render. In the same view, we included the segmentations as semi-transparent surface renders. These options allow us to combine the data and its analyzed interpretation in the same representation.

The expansion microscopy dataset (Fig. 1B) used here has multiple fluorescent channels representing microtubules, centrioles (marked by acetylated tubulin), and the nucleus. This dataset is challenging to visualize as the microtubule network is very dense,

and the difference in scale between the centriole and the cell is large. To emphasize the intricate network of microtubules, we used an isosurface render, where a surface mesh is generated at a single threshold value in the image. The threshold for the surface mesh can be interactively changed in Blender. By using the isosurface, the fibers in the microtubule signal are separated, and the differences in height are enhanced compared to a direct volumetric render. Similarly, the centrioles and the nucleus are also shown as isosurface renders. However, to emphasize the small centriole in this larger dataset, we used the interactively editable render properties to add a light-emission property to the centriole surface. Another way to emphasize small structures is to use Blender's extensive camera controls and animate the visualization properties of the different channels through animation time (Movie EV2).

With imaging at (or near) molecular resolution, it becomes increasingly important to visualize analyzed or inferred molecular models with the imaging data. To demonstrate this, we use a dataset acquired with U-Ex-STED (Woglar et al, 2022) (Ultra Expansion Microscopy combined with STED), which achieves molecular resolution of centrioles in RPE-1 cells. In this example, the U-Ex-STED signal for three of the different components of the centrioles are shown as light-emitting volumetric data, combined with procedural models of these proteins in the same view (Fig. 1C; Movie EV3). We use the "Geometry Nodes" system of Blender to programmatically construct these components: the microtubule (MT) wall is generated using the molecular structure of tubulin (with the procedural microtubule from (Andrea)), while the external torus of Cep63 and the internal γ-tubulin are represented as simplified spheres. This visualization method allows integrating analysis and incorporating theoretical models generated from different imaging modalities in the literature. For example, in the model shown, the MT barrel is generated by combining the atomic model of the centriole MT wall (Greenan et al, 2018), and the average sizes of the centriole barrel measured in EM tomograms (Guichard et al, 2013). This model combines data from multiple EM modalities, and fully captures the structures shown in the light-microscopy image obtained with U-Ex-STED super-resolution microscopy.

With Microscopy Nodes, we are able to load both light and electron microscopy data, changing only one load setting. To illustrate this, we use a manually segmented dinoflagellate phytoplankton volume EM stack, gathered with focused ion beam-scanning electron microscopy (FIB-SEM) (Mocaer et al, 2023) (Fig. 1D; Movie EV4). Here, we rendered the data as a volumetric density, which is light-scattering rather than light-emissive. This light-scattering render mode reflects the data acquisition of scanning electron microscopy, as this method captures the scattered electrons of the block surface. Additionally, we load the segmentation of the trichocysts (thread-like organelles that can be ejected) with Microscopy Nodes, which separates the segmentations for each organelle into separate surfaces and applies a colormap per object identity. The color can also be edited per object identity. Additionally, to show the trichocyst segmentation more clearly in the dense EM signal, we use two different slicing regions for the segmentation and the data, showing the trichocysts protruding from the EM density.

## Microscopy Nodes as a tool for effective 3D visualization

With the adaptability of the volumetric rendering of Blender powered by Microscopy Nodes, we are able to communicate new

biological results more effectively. To demonstrate this with a novel dataset (publicly available from https://uk1s3.embassy.ebi.ac.uk/idr/share/microscopynodes/20250730/FIBSEM_dino_masks.zarr), we present a late mitotic, dinoflagellate phytoplankton imaged with volume electron microscopy (FIB-SEM). Mitosis in most dino-flagellates is unique, and termed "dinomitosis" (Spector and Triemer, 1981; Drechsler and McAinsh, 2012). It is a form of closed mitosis, where the nuclear envelope remains intact and instead invaginates to form "nuclear tunnels". These tunnels allow the mitotic spindle to traverse the nucleus and assist with chromosomal segregation without entering the nucleoplasm (Gavelis et al, 2019).

FIB-SEM data is typically presented as single 2D slices of the data (Fig. 2A), where in this case, the nuclear tunnels appear as invaginations into the nuclear envelope within the slice plane. This visualization is also supported in Blender, allowing for arbitrary selections of viewing angles (Figs. 2B and EV1A) and setting the block as fully opaque. However, we can also utilize the advanced rendering engine to show these nuclear tunnels in 3D (Figs. 2C and Fig. EV1B), clipping the high and low ranges of EM intensities by making them transparent. Effectively, we only show electron densities filtered to the density corresponding to membranes. This approach allows us to look through the dense nucleus and visualize the 3D network of these nuclear tunnels in a single visualization. By clipping the high intensities, we remove the signal of the chromosomes, as they block the view of the nuclear tunnels. However, to highlight the position of the chromosomes, we add the masks of the chromosomes (manually segmented using Amira (Amira)) (Figs. 2D and EV1C). This visualization, showing both volumetrically rendered nuclear tunnels and semi-transparent label masks of the chromosomes, highlights how the tunnels may function to segregate the chromosomes while the nuclear envelope remains intact. Importantly, this visualization can be made with minimal adjustments to the default import parameters of the Microscopy Nodes interface.

Although many useful visualizations can be created with minimal adjustments to the default configuration, one of Blender's strengths is its highly flexible interface. By making more advanced changes to the shader setup, we can tailor the visualization specifically to this dataset. For example, the chromosomes in the dinoflagellate can also be added without manual segmentation by evaluating the data twice and displaying the EM densities corresponding to the chromosomes in a different color. This rendering mode can then show both structures of interest in the same visualization without manual segmentation (Fig. EV2A). In addition to rendering full or partial volumes, we can also render classic orthogonal slicing planes by duplicating the slicing cube and making the cubes very thin in one direction (Fig. EV2B).

As shown here, the best 3D representation of a dataset depends on the data that is visualized. Microscopy Nodes offers a single toolkit that allows for very versatile handling of a wide range of data types. Since it is built in the popular and extremely capable toolkit Blender, it also provides an extendable environment for customizing complex target visualizations.

## Design and implementation

Microscopy Nodes is a Blender extension written in Python and provides an interface to easily load any microscopy data, giving

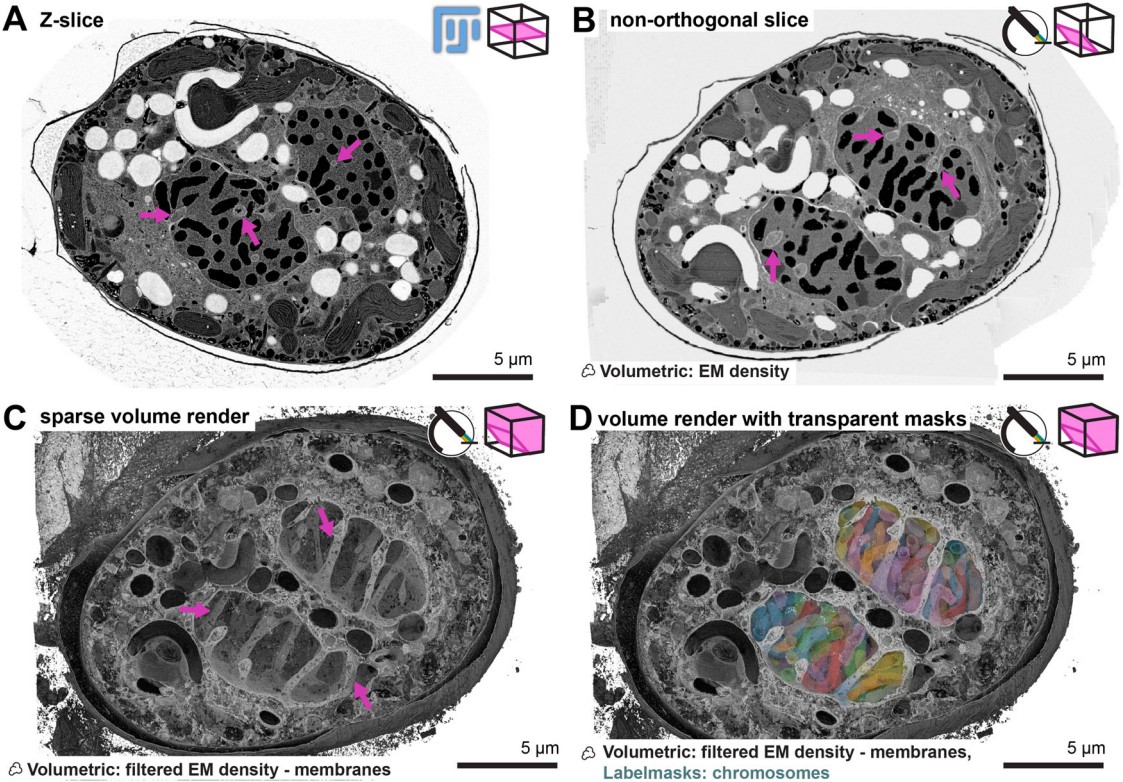

**Figure 2. Sparse 3D rendering assists in communicating new biological information.**

Different renders of the same dinoflagellate cell. Logos in the top right indicate the software used to render (Fiji or Microscopy Nodes), and a cartoon illustrating which data were used in the render (not to scale). Scale bars are shown as bars instead of grids as all images are orthographic projections. (A) Z-slice of a late mitotic dinoflagellate showing nuclear tunneling. A single slice of a FIB-SEM stack of a dinoflagellate showing invaginations in the nuclear envelope (magenta arrows), black is electron-dense matter. (B) A dense render of an arbitrary slicing axis in Blender of the same dataset. Here, the data were loaded into Blender with Microscopy Nodes to select a non-orthogonal slicing axis to show the nuclear tunnels (magenta arrows). (C) Rendering some EM densities as transparent can give a 3D overview of biological structures. A volumetric render in Blender, highlighting low EM densities as dense volumetric renders to highlight low-density structures such as membranes. This visualization shows the 3D network of tunnels in the nucleus (magenta arrows) in a single rendered image. (D) Blender can combine semi-transparent masks with volumetric rendering. A 3D render showing sparse volumetric data of membranes, combined with semi-transparent multicolored chromosome masks. Each separate chromosome has a separate color (of a revolving colormap of ten colors).

choices between preset load settings. The user can then interactively use the extensive Blender interface to edit, annotate, animate their data, and render the final output image either from the interface or the command line (Fig. 3A). Microscopy Nodes loads microscopy data, saves the data into Blender-compatible files (namely VDB (Museth, 2013) volume files and alembic (alembic) mesh files), loads these files into Blender, and applies presets and Microscopy Nodes-specific user settings.

Microscopy data brings along large data sizes, diverse file formats and saving techniques. Blender currently has a notable limitation that its default "quick" rasterized rendering engines (such as 'EEVEE', but also the viewport "Surface"' and "Wireframe" modes) do not support more than 4 Gibibytes (GiB) of volumetric data. The raytracing render mode "Cycles", however, can handle large volumetric data. To allow users with large data to flexibly use Microscopy Nodes, we implemented a reloading scheme, where one first loads a smaller version of the data (under 4 GiB per timeframe for all loaded channels combined)—and only upon final render in Cycles, exchange it for the full/larger scale copy (Fig. 3A). This downscaling of data offers additional benefits as it allows for fast adjustment of the render settings on e.g., a personal computer

which can eventually be transferred to a larger workstation or HPC cluster for the final render at full resolution. This feature is critical as working in Cycles with larger files requires sufficient RAM to fit the (temporary) VDB files comfortably. For example, multiple figures in this manuscript were made on a 32GB RAM M1 Macbook Pro (Figs. 1A,D, 2A–D and EV2A,B; Movie EV1), but for larger data or long movies, the movies were made on workstations or prepared on a laptop and then transferred to an HPC cluster for final rendering.

Microscopy Nodes supports both Tif and OME-Zarr (Moore et al, 2023) files. For working at smaller scales, Microscopy Nodes utilizes the presaved multi-scales of an OME-Zarr file or dynamically down-samples data from a Tif file to fit within the 4 GiB limit (Figs. 3B, inset 1 and EV3). It can handle 8 bit to 32 bit integer and floating point data, although all data types will be resaved into 32 bit floating point VDB files, which can cause temporary files to take up more space than the original. Microscopy Nodes loads 2D to 5D files containing data across time, z, y, x and channels, in arbitrary order (can be remapped in the user interface as well, Fig. 3B, inset 2). To focus on relevant data, users can clip the time axis, which can be useful for long videos.

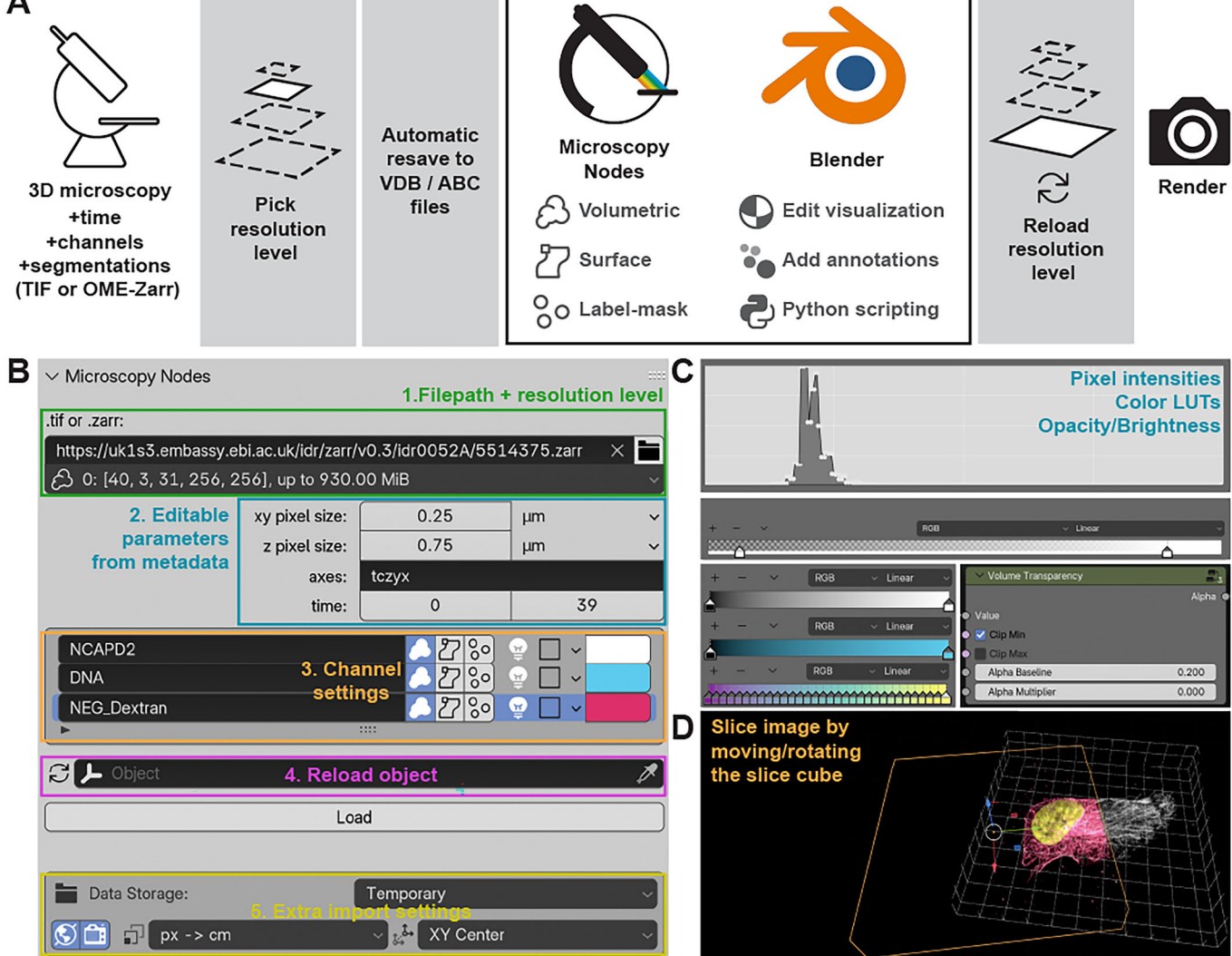

**Figure 3. User interface of Microscopy Nodes.**

(A) The Microscopy Nodes workflow allows for interactive use of big data. Up to 5D microscopy files with segmentation channels can be loaded into the Microscopy Nodes add-on, which resaves the data into Blender-appropriate formats. The Microscopy Nodes Blender interface allows for adjusting settings for each channel and interactive editing. A smaller dataset can be used interactively, to then up-scale for the final render. Icons used are from the Blender icon library. (B) A variety of image features are readily specified in the Microscopy Nodes panel in Blender. (1) Input is specified as a path to a Tif or OME-Zarr file, or using a URL linking to an OME-Zarr file in a repository. This provides a selection of presaved or downsampled resolutions (Fig. EV3) for large data. (2) Pixel sizes, axis order, and number of frames are read out automatically, but they can also be specified or edited manually. (3) For each channel, users can select (from left to right): load as volumetric, load as isosurface, load as integer label mask, whether the data should emit or absorb light, colormap (choice from a single color, or a short list of nonlinear defaults), and color of the channel (in single color mode) respectively. If specified, Microscopy Nodes will also read out channel names from the metadata. (4) Users can reload a previous volume to change the underlying data or data resolution, and overwrite loading variables by pointing this field to a previously loaded dataset. (5) By default, Microscopy Nodes will store local files in a temporary folder, but this location can also be specified by the user. Additionally, this section contains buttons specifying whether to reset the background color and render settings upon loading, and details of the coordinate space to load in, with the options (px -> cm, Å -> m, nm -> m, up to m -> m). The bottom right button changes the import location; the default is centered in the XY plane. Changing this may make it easier to align externally created meshes/annotations. All buttons have hover-tooltips in the software. (C) Lookup tables can be interactively remapped with a histogram of the voxel values. Microscopy Nodes has a histogram and accompanying pixel intensity range sliders, that affect a color LUT, which can be remapped to one of hundreds of defaults (pyapp-kit/cmap, 2025) upon right click, or recolored with a color picker. Transparency is handled by a separate box, and will define the brightness for the emissive signal, and the opacity of the non-emissive signal. This has a baseline constant value, and linear response signal multiplier, and can optionally clip the out-of-range values to be transparent. (D) Slicing a volume can be achieved by moving the slice cube in the 3D space. To illustrate slicing, two copies of the same dataset were loaded, where one is sliced with the selected slice cube (orange, sliced data in magenta), and the other is shown unsliced in white.

Biological images often include multiple channels with varying data, where each channel should be visually distinct to aid interpretation. To simplify the setup of distinct visualization styles for different channels, Microscopy Nodes provides a dedicated channel interface (Fig. 3B, inset 3). This interface allows editing channel names, loading as volumetric, surface or label-mask, and changing material properties, such as color and emission. For each channel, the user gets a color picker to set the color, or the option to switch to one of a few preset colormaps (Cmap). To allow for easier loading and custom workflows, default settings for loading can be set in the add-on preferences (Fig. EV4).

To ensure a smooth default user experience, Microscopy Nodes maps every pixel to a centimeter in Blender-space (Blender works best near the meter length scale) and rescales the z axis to isotropic. However, for users who need to align data from other workflows, alternative input transforms are available. These include options to map each micrometer, nanometer, or Ångström to a Blender-meter, as well as settings to adjust input positioning, such as centering the data in the scene or aligning it to the scene axes to facilitate easier colocalization (Fig. 3B, inset 5).

After loading, the user can interactively edit the data visualization by adjusting the intensity, the transparency and brightness of the object, and changing colors (Fig. 3C). Data can be sliced interactively by moving and rotating one or multiple slice cubes (Figs. 3D and EV2B). In addition to these default Microscopy Nodes adjustments, Blender provides visualization control down to a granular detail of how light in the scene interacts with the volumetric data. Microscopy Nodes provides an extensible and versatile environment for 3D microscopy visualization, while also providing useful presets and easy and intuitive tools.

By default, the Microscopy Nodes toolkit only uses a small segment of the extensive Blender user interface. However, the vast rest of the interface remains accessible and can be used as needed. As such, Blender allows for, e.g., the drawing of annotations manually through the many modeling tools or programmatic creation and animation of meshes, either from its embedded graphical programming language or Python scripting. Alignment and scaling of meshes can also be improved when importing to a physical world scale (e.g., mapping μm to Blender-meter). Blender can also be further extended with other (scientific) add-ons, such as Molecular Nodes (Johnston) or Mastodon cell-tracking (mastodon).

## Blender as a platform for microscopy

With the great versatility in both input modality and target visualization, we believe Microscopy Nodes has the potential to revolutionize the visualization of 3D microscopy data, as it makes Blender, a comprehensive rendering and modeling tool, accessible for all labs and any biologist, regardless of their computational experience. This paves the way for Blender to act as a platform for an ecosystem of tools for microscopy data visualization and analysis.

Currently, Microscopy Nodes purely serves a visualization purpose, and the core of the add-on is loading data that is easily adaptable, rather than providing analysis tools. However, Microscopy Nodes can already be used as a loading interface for mesh analysis in Blender (Claussen), and once volumetric math and mesh analysis in Blender mature, we anticipate that new image analysis tools will be incorporated or built on top of Microscopy Nodes.

Although several tools for 3D visualization of bioimages already exist and offer essential features for microscopy data (Table 1), many are proprietary, and open-source alternatives often struggle to deliver a comprehensive user experience, such as advanced animation and annotation controls. Proprietary solutions may offer some of these capabilities, but they are frequently limited by licensing costs, platform restrictions, and a lack of customizability. In contrast, Blender is a mature, well-supported open-source platform with a large community of developers that excels in both animation and visualization. By integrating microscopy-specific functionality through Microscopy Nodes, Blender becomes a uniquely powerful solution that bridges the gap between high-end graphics capabilities and the specialized needs of bioimage visualization.

Microscopy Nodes is available for download directly from the Blender interface, the Blender extensions platform, and GitHub. It is documented on https://aafkegros.github.io/MicroscopyNodes/ and through a series of YouTube tutorials, which are kept up to date with Blender and Microscopy Nodes updates. Visualizations in this publication were made with Blender 4.4 and Microscopy Nodes 2.2.

## Methods

### Reagents and tools table

| Reagent/resource | Reference or source | Identifier or Catalog Number |
|---|---|---|
| **Experimental models** | | |
| hTERT-RPE1 | Atorino ES, Hata S, Funaya C, Neuner A and Schiebel E (2020) CEP44 ensures the formation of a bona fide centriole wall, a requirement for the centriole-to-centrosome conversion. Nat Commun 11: 903 Cell line obtained from Schiebel lab, mycoplasma tested | |
| **Antibodies** | | |
| α-Tubulin | Geneva Antibody Facility | ABCD_AA344 |
| β-Tubulin | Geneva Antibody Facility | ABCD_AA345 |
| Acetyl-α-Tubulin | Thermo Fisher Scientific | 32-2700 |
| γ-Tubulin | Proteintech | 15176-1-AP |
| CEP63 | Abcam | ab235513 |
| Donkey anti-Guinea pig, Alexa Fluor 488 | Jackson ImmunoResearch | 706-545-148 |
| Donkey anti-mouse, Alexa Fluor 594 | Thermo Fisher Scientific | A-21203 |
| Goat anti-mouse, Abberior STAR 635 P | Abberior | ST635P-1001-500 μg |
| **Chemicals, enzymes and other reagents** | | |
| Methanol | VWR International GmbH | 1.06009.1000 |
| Acrylamide | Sigma Aldrich | A4058-100mL |
| Formaldehyde solution | Sigma Aldrich | F8775 |
| Sodium acrylate | Sigma Aldrich | 408220 |
| *N,N'*-methylenebisacrylamide (Bis) | Sigma Aldrich | M1533 |
| TEMED | Thermo Fisher Scientific | 17919 |
| Ammonium persulfate (APS) | Thermo Fisher Scientific | 17874 |
| Tris base | Sigma Aldrich | T1503 |

| Reagent/resource | Reference or source | Identifier or Catalog Number |
|---|---|---|
| SDS | Serva | 20765.01 |
| Sodium chloride (NaCl) | VWR International GmbH | 1.06404.1000 |
| Poly-D-lysine | Merck | A-003-E |
| BSA | Cell signaling | 9998S |
| Triton X-100 | Sigma Aldrich | X100-1L |
| Tween-20 | Promega | H5151 |
| Hoechst nuclear dye | Thermo | H1399 |
| 1-Hexadecene | Sigma Aldrich | 822064 |
| Glass-distilled acetone | EMS | 10015 |
| Uranyl acetate | Agar | R1260A |
| Crosslinker D Lowicryl HM20 | Polysciences Inc. | 15924B |
| Monomer E Lowicryl HM20 | Polysciences Inc. | 15924 A |
| EM-Tec AG44 conductive silver paint | Micro to Nano | 15-002144 |
| **Software** | | |
| Zeiss Zen (Blue edition) 3.12 | | |
| Leica LASX 5.2.2 | | |
| Zeiss SmartSEM | | |
| Zeiss Atlas 5 | | |
| ImageJ 1.54p | | |
| Amira 3D 2023.2 (Thermo Fisher Scientific) | | |
| Blender 4.4 | | |
| Dask 2025.5.1 | | |
| Tifffile 2025.6.11 | | |
| imagecodecs 2025.3.30 | | |
| zarr-python 3.0.8 | | |
| cmap 0.6.0 | | |
| s3fs 2025.5.1 | | |
| zmesh 1.8.0 | | |
| databpy 0.3.0 | | |
| **Other** | | |
| 12 mm round #1.5 coverslips | Fisher Scientific GMBH | 11846933/ CB00120RAC20MNZ0 |
| Zeiss LSM 980 AIRY Fast | | |
| Leica LASX Stellaris 8 STED | | |
| Leica EM-ICE | | |
| Leica EM-AFS2 | | |
| Zeiss LSM 780 NLO | | |
| Sputter coater Quorum Q150RS | | |
| Zeiss Crossbeam 540 | | |

## Electron microscopy

### Sample collection and cryo-fixation

- As part of the Traversing European Coastlines (TREC) expedition (https://www.embl.org/about/info/trec/), environmental phytoplankton communities were collected near the Kristineberg Centre for Marine Research and Innovation using a 10 µm plankton net. The sample was fractionated using a 40 µm sieve.
- Using the Advanced Mobile Laboratory (AML, https://www.embl.org/groups/mobile-labs/) at the marine station, the sample was concentrated on a 1.2 µm mixed cellulose ester membrane and allowed to sediment before high-pressure freezing using the EM-ICE system (Leica Microsystems).
- A 1.2 µl drop of the concentrated phytoplankton mix was loaded into a hexadecene-coated Type A carrier (gold-coated copper; 3 mm wide × 200 µm deep). The sample was then sandwiched with a Type B aluminium carrier before freezing.

### Freeze substitution and resin infiltration

Freeze substitution was performed in an EM-AFS2 system (Leica Microsystems) using previously published protocols (Mocaer et al, 2023; Ronchi et al, 2021)

- Samples were incubated in 0.1% uranyl acetate in dry acetone at −90 °C for 72 h.
- The temperature was raised to −45 °C at a rate of 2 °C per hour, followed by incubation at −45 °C for an additional 10 h.
- Samples were infiltrated with Lowicryl HM20 resin at increasing concentrations (10, 25, 50, 75, 100%) while raising the temperature to −25 °C.
- This was followed by three exchanges with 100% Lowicryl resin, each lasting 10 h.
- UV polymerization was carried out at −25 °C for 48 h, then continued at 20 °C for another 48 h.

### Confocal imaging and targeting

- A 3D confocal map of the resin block was generated using a Zeiss LSM 780 NLO microscope.
- The block was mounted upside down on a drop of water in a glass-bottomed round dish.
- A 4 × 4 tiled scan was acquired with a 25X/0.8 NA multi-immersion objective.
- The cell of interest was identified and branded using a 2-photon laser set at 800 nm and 10% laser power.

### SEM preparation

- The block was mounted onto a scanning electron microscopy (SEM) stub using superglue.
- Silver paint was applied around the sample area to enhance conductivity.
- The stub was gold-sputtered at 30 mA for 180 s using a Q150RS coater (Quorum).

### FIB-SEM acquisition

Whole-cell volume imaging was performed using a Zeiss Crossbeam 540 focused ion beam-scanning electron microscope (FIB-SEM) at an isotropic voxel size of 10 nm.

- Milling was conducted at 30 kV and 3 nA.
- SEM imaging was carried out at an accelerating voltage of 1.5 kV and a beam current of 700 pA, using an ESB detector (ESB Grid set to 1110 V).

**Table 1.** Comparison of features of 3D bioimage visualization tools.

| | Microscopy Nodes (Blender) | Amira (Thermo Fisher) | Imaris (Oxford Instruments) | Arivis (Zeiss) | Agave | BigVolumeViewer (FIJI) | Sciview (FIJI) |
|---|---|---|---|---|---|---|---|
| Free and open-source software | ✓ | ✗ | ✗ | ✗ | ✓ | ✓ | ✓ |
| Raytraced volumes | ✓ | ✗ | ✗ | ✗ | ✓ | ✗ | ✗ |
| Voxel and volume analysis | ✗ (In progress) | ✓ | ✓ | ✓ | ✗ | (Not integrated) | (Not integrated) |
| Mesh analysis and editing | ✓ | ✗ (Only analysis) | ✗ (Only analysis) | ✗ (Only analysis) | ✗ | ✗ | ✗ |
| Multi-resolution data | ✓ (Static) | ✓ (Requires additional software) | ✓ (Dynamic) | ✓ (Dynamic) | ✓ (Static) | ✓ (Dynamic) | ✗ |

Checkmarks mean this is provided, crosses mean this is not present. **Free and open-source software** refers to any openly licensed software, giving access to source code and not requiring payment. **Raytraced volumes** refer to an engine that uses physical light scattering to render. **Voxel and volume analysis** mean any software that can arbitrarily use voxels to calculate new metrics from the graphical user interface. While Microscopy Nodes does have the application of isosurface thresholds, it does not yet have full voxel analysis capabilities. The FIJI plugin visualization toolkits are noted as "(not integrated)" as FIJI has many volume analysis tools, but they are not directly accessible from the volume viewing interface. **Mesh analysis and editing** refers to the ability to measure meshes (volume, curvature, etc.) but also to the ability to sculpt, generate and edit meshes. **Multi-resolution data** refers to the ability to load from a multi-resolution data source, such as OME-Zarr, and switch between scales of the data. Static refers to static copies of the resolution that the users have to switch out explicitly, while dynamic refers to automatic resampling to adapt the image resolution to the view.

- The final dwell time per pixel during acquisition was 11 μs.
- Raw image stacks were aligned using the Linear Stack Alignment with SIFT plugin in Fiji.
- Individual chromosomes were segmented using Amira (Thermo Fisher Scientific) via intensity thresholding.

## Expansion microscopy

### Sample Preparation
Adapted from Gambarotto et al, (2019).

A. Cell preparation and fixation

- Culture hTERT-RPE-1 cells under standard conditions and seed on 12 mm round coverslips.
- Fix 5 min in pre-chilled ($-20\,°C$) methanol; keep samples cold and do not exceed 5 min.

B. Anchoring

- Incubate coverslips in 2% acrylamide/1.4% formaldehyde in PBS for 3 h 30 min at 37 °C, fully covering samples and preventing evaporation.

C. Monomer infiltration and gelation

- Prepare monomer in PBS: 23% (w/v) sodium acrylate, 10% (w/v) acrylamide, 0.1% (w/v) $N,N'$-methylenebisacrylamide, 0.5% TEMED, 0.5% APS.
- Add APS and TEMED last, mix quickly, minimize oxygen exposure, and incubate 1 h at 37 °C in a dark, humidified chamber to polymerize.

D. Homogenization/denaturation and first expansion

- Place gels in denaturation buffer (50 mM Tris base, 200 mM SDS, 200 mM NaCl, pH 9) and rock 15–30 min at room temperature.
- Continue denaturation 1 h 30 min at 95 °C in a thermomixer (same buffer), handling gels gently with wide tools.
- Expand overnight in ddH$_2$O at room temperature, replacing water several times until size stabilizes; keep gels fully submerged.

E. Post-expansion immunostaining
1. Equilibrate and block

- Incubate gels in PBS, 1 h at room temperature.
- Block 30 min at 37 °C in 3% BSA + 0.05% Triton X-100 in PBS.

2. Primary antibodies (overnight, 4 °C, gentle agitation; dilutions for post-expansion staining)

- For Fig. 1B: α-Tubulin 1:1500 (Geneva Antibody Facility, ABCD_AA344); β-Tubulin 1:1500 (Geneva Antibody Facility, ABCD_AA345); Acetyl-α-Tubulin (Lys40) 1:500 (Thermo Fisher, 32-2700).
- For Fig. 1C: γ-tubulin 1:500 (Proteintech, 15176-1-AP); acetyl-α-tubulin (Lys40) 1:500 (Thermo Fisher, 32-2700); CEP63 1:500 (Abcam, ab235513).

3. Wash

- 3 × 30 min in PBS + 0.1% Tween-20 at room temperature.

4. Secondary antibodies (2 h 30 min at 37 °C, gentle agitation)

- For Fig. 1B: Donkey anti-Guinea pig Alexa Fluor 488, 1:1000 (Jackson, 706-545-148); Donkey anti-Mouse Alexa Fluor 594, 1:1000 (Thermo, A-21203).
- For Fig. 1C: Same as above, plus Goat anti-Mouse Abberior STAR 635P, 1:1000 (Abberior, ST635P-1001-500 μg).

5. Nuclear counterstain

- Hoechst, 10 min at room temperature.

6. Final washes and expansion

- Wash 3× in PBS + 0.1% Tween-20, then expand 1 h in ddH$_2$O at room temperature (replace water once); avoid drying or stretching gels.

F. Imaging hand-off

- Transfer fully expanded gels to imaging dishes or poly-D-lysine–coated #1.5H precision coverslips; minimize compression by using spacers if adding a top coverslip.

G. Typical timing

- Day 1: Fixation (≤10 min), anchoring (~3.5 h), gelation (~1 h), denaturation (~2 h), start overnight expansion.
- Day 2: PBS equilibration and blocking (~1.5 h), primary antibody incubation (overnight).

- Day 3: Washes and secondary (~3.5 h), Hoechst (10 min), final expansion (1 h), imaging.

### Confocal and super-resolution imaging

A. Mounting of expanded gels

- Coat 25 mm round precision coverslips (#1.5H) with poly-D-lysine according to the manufacturer's instructions; rinse and air-dry.
- Trim expanded gels into small pieces and place them onto the coated coverslips; remove excess buffer without compressing the gel.

B. Confocal imaging for Fig. 1B (Zeiss LSM 980 AIRY Fast)

- Microscope and optics: Zeiss LSM 980 with AIRY Fast; 40×/1.2 NA water-immersion objective.
- Acquisition mode: xyz confocal scanning; adjust voxel size and z-step to satisfy Nyquist for the expanded specimen.
- Immersion medium: water; maintain a stable water column during acquisition.
- Detector and channels: set according to fluorophore(s) used in Fig. 1B; match laser lines and emission windows accordingly.
- Image stacks: acquire fields of interest under identical conditions across replicates for quantitative comparison.

C. Confocal and STED imaging for Fig. 1C (Leica STELLARIS 8 STED)

- Microscope and optics: Leica STELLARIS 8 STED with white light laser (WLL) and pulsed STED depletion lasers at 589, 660, and 775 nm; HC PL APO CS2 86×/1.20 WATER objective; immersion medium water ($n = 1.33$).
- Scan settings: xyz mode; bidirectional scanning at 300 Hz; zoom factor 16.65.
    C1. Confocal channel for γ-tubulin
- Excitation: 499 nm from WLL at 2% laser power.
- Emission detection: 430–506 nm using HyD S detector in photon counting mode.
- Pinhole: 91.2 μm (599.68 mAU).
- Pixel dwell time: 7.04 μs.
- Sampling and volume: 200 × 200 pixels; voxel size 0.041 μm (X–Y) × 0.257 μm (Z); 15 Z-planes; field of view 8.12 × 8.12 μm (X–Y); Z-extent 3.6 μm.
    C2. STED channels for acetylated tubulin and CEP63
- Excitation: WLL set to the appropriate lines for each fluorophore.
- Depletion: pulsed STED at 589 nm and 775 nm at 100% depletion power.
- Emission detection: HyD X detectors; 601–764 nm window.
- τ-STED lifetime gating: 0.3–8 ns enabled to enhance resolution and suppress background.

- Sampling and volume: $368 \times 368$ pixels; pixel size $0.022 \,\mu m$ (X–Y); Z-step $0.225 \,\mu m$; 17 Z-planes; field of view $8.12 \times 8.12 \,\mu m$ (X–Y); Z-extent $3.6 \,\mu m$.

D. General notes for both setups

- Keep gels stably adhered to the substrate; minimize drift during long Z-stacks.
- Use consistent detector gain, laser power, and gating across replicates intended for comparison.
- Record all acquisition metadata (objective, NA, voxel size, dwell time, pinhole, depletion power, gating window) with each dataset.

## Data availability

Microscopy Nodes is open source and published under a GPL-3 license, hosted at https://github.com/aafkegros/MicroscopyNodes, and directly installable from the Blender extension panel. Newly acquired microscopy images visualized in this study were deposited in the IDR repository (Williams et al, 2017), under the entries: https://uk1s3.embassy.ebi.ac.uk/idr/share/microscopynodes/20250730/FIBSEM_dino_masks.zarr. https://uk1s3.embassy.ebi.ac.uk/idr/share/microscopynodes/20250730/RPE1_4x.zarr. https://uk1s3.embassy.ebi.ac.uk/idr/share/microscopynodes/20250730/UExSTED.zarr. Previously archived public datasets are referenced in the text.

The source data of this paper are collected in the following database record: biostudies:S-SCDT-10_1038-S44319-025-00654-8.

## Peer review information

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

## Acknowledgements

We thank Eugene Katrukha for help with BigVolumeViewer, Nemo Andrea for his microtubule model, Samuel Pantze for helping with Blender bug finding, and the advanced light-microscopy facility (ALMF), the advanced mobile laboratory (AML) and the electron microscopy core facility (EMCF) at the European Molecular Biology Laboratory (EMBL) in Heidelberg, Evident/Olympus, and Zeiss for their support; IT and HPC resources at the EMBL in Heidelberg for providing the essential computational infrastructure and the Imaging Centre at the European Molecular Biology Laboratory (EMBL IC) for access to infrastructure and services, supported by the Boehringer Ingelheim Foundation. We thank Jan Ellenberg for his feedback on our manuscript. For the support with environmental sampling, we thank the Traversing European Coastlines (TREC) Consortium, the TREC core partners EMBL, the Tara Ocean Foundation, the Tara Europa Consortium and the European Marine Biological Resource Centre (EMBRC) for their commitment to making the TREC expedition possible, and especially the Kristineberg Centre for Marine Research and Innovation. This work was supported by the European Molecular Biology Laboratory and by the Deutsche Forschungsgemeinschaft (DFG, German Research Foundation) - project number 452616889.

## Author contributions

**Aafke Gros**: Conceptualization; Software; Formal analysis; Visualization; Methodology; Writing—original draft; Writing—review and editing. **Chandni Bhickta**: Data curation; Investigation; Writing—review and editing. **Granita Lokaj**: Data curation; Investigation; Writing—review and editing. **Brady Johnston**: Software; Writing—review and editing. **Yannick Schwab**: Resources; Supervision; Writing—review and editing. **Simone Köhler**: Supervision; Funding acquisition; Validation; Writing—original draft; Writing—review and editing. **Niccolò Banterle**: Conceptualization; Supervision; Validation; Visualization; Writing—original draft; Writing—review and editing.

Source data underlying figure panels in this paper may have individual authorship assigned. Where available, figure panel/source data authorship is listed in the following database record: biostudies:S-SCDT-10_1038-S44319-025-00654-8.

## Funding

## Disclosure and competing interests statement

The authors declare no competing interests.

# Expanded View Figures

**Figure EV1.   Shader configuration in Fig. 2.**

A repeat of the figure panel of Fig. 2 and accompanying a screenshot of the shader configurations. Data were loaded from https://uk1s3.embassy.ebi.ac.uk/idr/share/microscopynodes/FIBSEM_dino_masks.zarr with emission off, with the data channel as volumetric, and the chromosomes as label mask. All changes made from the default Microscopy Nodes shader settings are outlined in magenta. (**A**) The dense orthographic render of Fig. 2B is made by setting "Alpha Baseline" high (here 10), making the block very dense, and turning "Clip Min" and "Clip Max" both off, showing the maximum and minimum values as dense black and white, instead of transparent. Additionally, we adjusted the pixel intensities to show the nuclear tunnels and inverted the colormap. The inverted colormap can either be achieved by right-clicking the colormap box, or be set as the default in the add-on preferences. (**B**) To generate the sparse volume render, the pixel intensity window that is selected is a window that corresponds to the EM densities of membranes, "Clip Min" and "Clip Max" are ticked, rendering all data outside of the pixel intensity window as transparent. Additionally, this render uses high settings for the scattering of light in the render (Render Properties > Volumes > Max Steps) and how many times a ray of light bounces (Render Properties > Light Paths > Transparent/Total/Volume). This allows the cavities to be clearly darker as light scatters inside it. Here, other settings, such as scattering anisotropy, defining whether light scatters more forward or backward, have been left unchanged. (**C**) The same render as in (**B**), with included masks. These are loaded from a channel of the dataset containing label masks for the chromosomes (bitmap images that contain separate values for each object), and loaded as label mask with emission off and the colormap Tab10 (this can be selected from the loading window, or by right clicking on the colormap here), the only change made is to reduce the opacity to 0.2, to allow us to still see the nuclear tunnels.

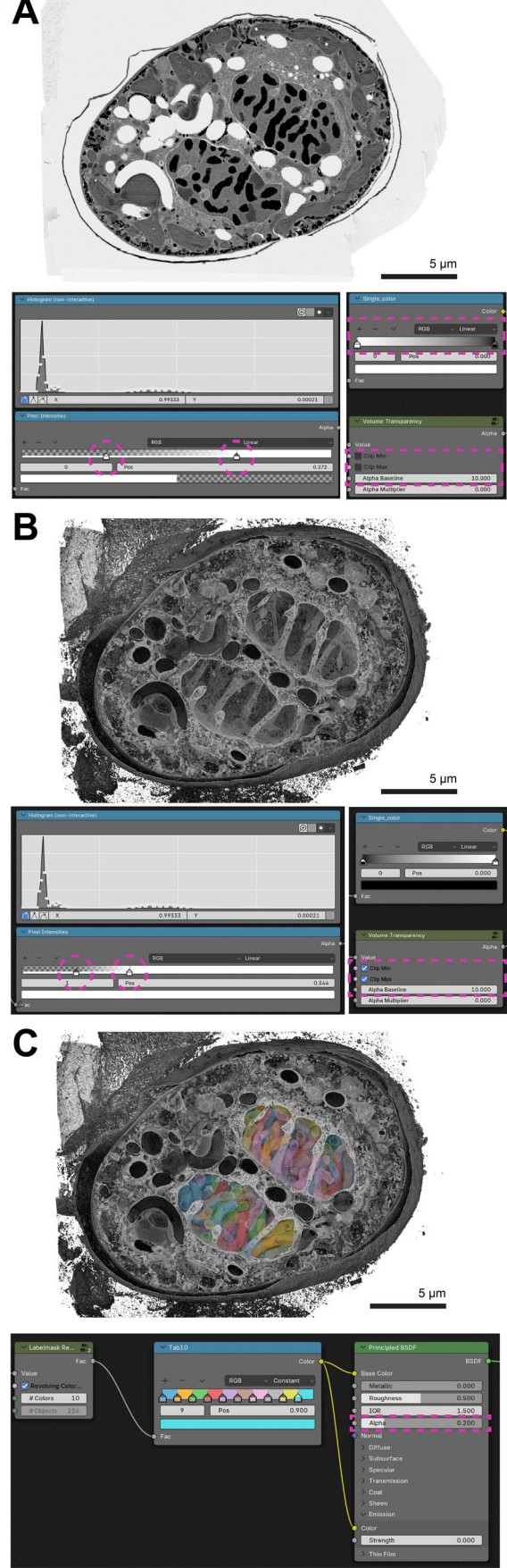

**A**

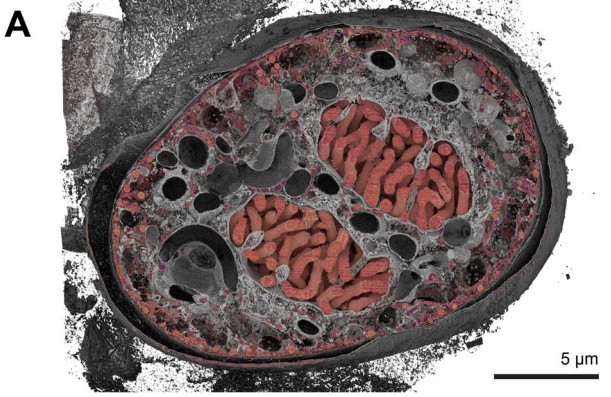

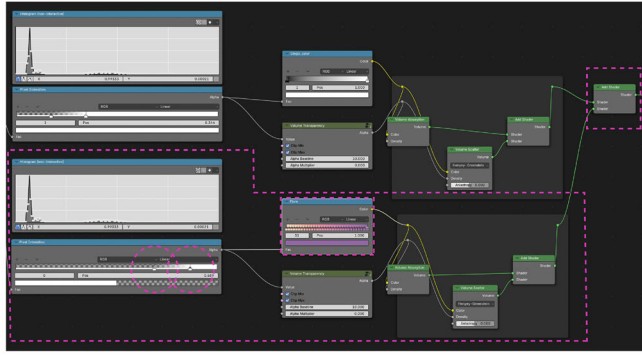

**B**

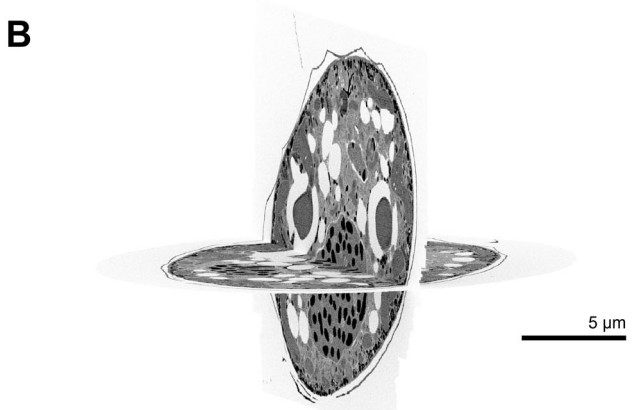

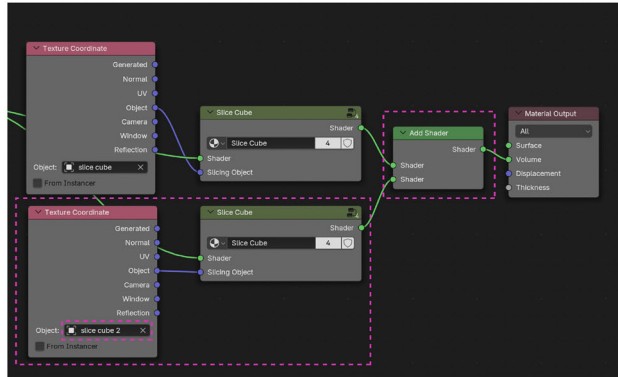

◀

**Figure EV2.** **The configuration of visualization modes is very flexible.**

(**A**) Structures with different EM densities can also be displayed with separate colormaps in the same visualization. This is done by duplicating the parsing of data. Here, membranes are visualized in gray (settings as described in Fig. EV1B), and chromosomes are visualized by selecting the appropriate (higher) pixel intensity window and adjusting the colormap (magenta boxes). The colormap for this region is set to seaborn::flare, which can be found under the right click on the lookup table. (**B**) This is a shader setup to show how to add a second slicing cube. This uses the same visualization as used in Figs. 2B and EV1A. Here, the normal slicing cube is made very thin in one axis to make it into a single slicing plane, and copied to make a second slicing cube object. In the shader of the volume, we copied the slicing mechanism and pointed it to the new cube. We combine the two slices with an "Add Shader" node.

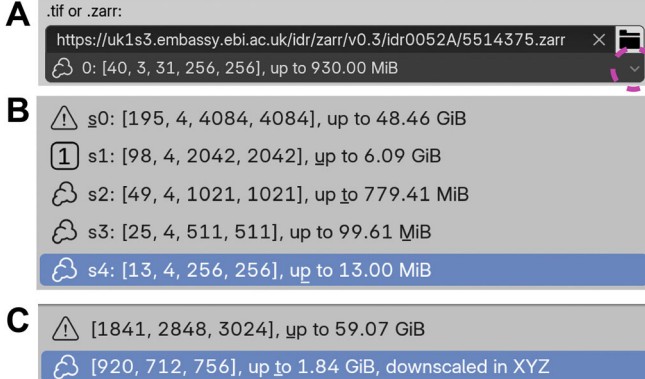

**Figure EV3. Selection of data scales allows switching between computational complexities.**

(A) The location of the scale selection window, directly under the selected path, the button to show all scale options is circled in magenta. (B) Selection window for an OME-Zarr with presaved scales. The warning icon is for scales that only work in the raytraced renderer, the 1 icon is if only one channel fits in the rasterized rendering. (C) Selection window for Tif scales. Smaller scales than 4 GiB per frame are dynamically generated by downsampling. The algorithm prioritizes downsampling in Z (twofold), then does XY downsampling (fourfold), until the dataset is less than 4 GiB per frame.

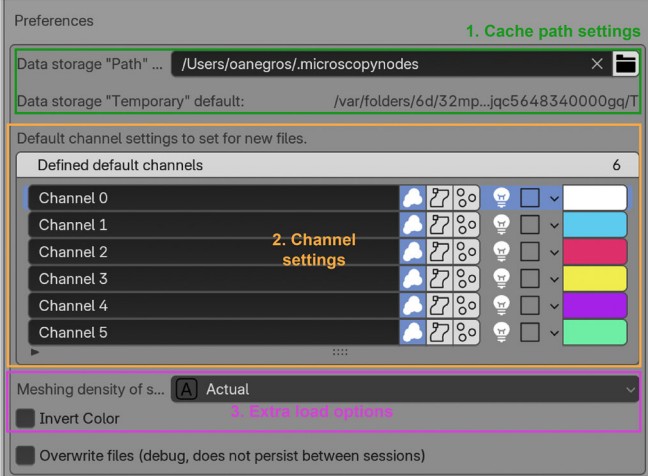

**Figure EV4. Default load settings can be configured in preferences.**

The preferences can be found under Edit > Preferences > Add-ons > Microscopy Nodes. (1) This allows showing the default static path for local storage, and the auto-generated temporary path used. These are used when the "Data Storage" in the main Microscopy Nodes is set to "Path" and "Temporary", respectively. (2) The default channel settings are used whenever a new filepath or URL is entered. This allows users to set up their Microscopy Nodes environment to their usual use cases, including colors, or how data is organized. (3) Extra load options comprise the default mesh density of masks and surfaces (default matches the vertex density to the pixel size, but coarser meshes can be selected), and an option to invert color by default.

