## [Peer Review File · EMBO Reports]

Microscopy Nodes: versatile 3D microscopy visualization with Blender

Aafke Gros, Chandni Bhickta, Granita Lokaj, Brady Johnston, Yannick Schwab, Simone Köhler, and Niccolo Banterle

Corresponding author(s): Niccolo Banterle (niccolo.banterle@embl.de) , Aafke Gros (aafke.gros@embl.de)

Review Timeline:

Transfer Date:	7th Aug 25
Editorial Decision:	30th Sep 25
Revision Received:	27th Oct 25
Accepted:	12th Nov 25

Editor: Esther Schnapp

Transaction Report: This manuscript was transferred to EMBO reports following peer review at Review Commons.

Review
COMMONS

Review #1

1. Evidence, reproducibility and clarity:

Evidence, reproducibility and clarity (Required)

The work by Gros et al. presents a paper introducing Microscopy Nodes, a new plugin for Blender 3D visualization software designed to import and visualize multi-dimensional (up to 5D) light and electron microscopy datasets. Given that Blender is not directly suited for such tasks, this plugin significantly simplifies the process, making its visualization engine accessible to a wide range of researchers without prior knowledge of Blender. The plugin supports importing volumes and labels from generic TIF or modern OME-Zarr image formats and includes supplementary video tutorials on YouTube to facilitate basic understanding of the visualization workflows.

****Major comments:****

- The manuscript suggests that Microscopy Nodes can easily handle large datasets, as evidenced by the showcases. However, in my personal tests, I was unable to import a moderate TIF stack of about 5GB, which is considerably smaller than the showcased datasets. Post-import, a data cube was displayed, but the Blender interface became unresponsive. The manuscript should include a section stating limitations and addressing issues and providing suggestions for visualization of large datasets.
- The feature of importing Zarr-datasets over HTTP is great, but the import process was very slow in my tests, even on a robust network. For reference, loading 1.8 GB of the PRPE1_4x dataset at s1 level took 52 minutes. This raises concerns about potential code issues and general usability of the suggested workflow.
- The onsite documentation is a bit outdated and fails to fully describe the plugin settings.
- The YouTube tutorials feature an outdated version of the plugin, which could confuse the general microscopy audience. These should be updated to better align with the current plugin functionality. Additionally, using smaller, easily accessible datasets for these tutorials would improve user testing experiences. Hosting complete (downsampled) demo project folder on platforms like zenodo.org could also enhance usability of such tutorials.
- The manuscript describes a novel dataset used in Fig. 2, but no reference is provided. Additionally, practical implementation of the coloring description for Fig. 2D can be unclear for inexperienced users, necessitating either step-by-step instructions or the provision of downsampled Blender files to aid understanding.

[OPTIONAL] When importing labels, they can be assigned to individual materials only if

initially split into multiple color channels. It would be great if the same logic is implemented when those materials are provided as indices within a single color channel. There can be a switch to define the logic used during the import process: e.g. the current one, when the objects are just colored based on a color map, or when they are arranged as individual materials as done when labels are imported from multiple color channels.

****Minor comments:****

- The manuscript shows nice visualizations of time series, light, and electron microscopy datasets, but in its current state, it is targeted more for light microscopy, where the signal is white. On the other hand, many EM datasets are rendered in inverted contrast (TEM-like), where the signal is black. To render such volume properly, it is needed to go into the Shading tab and flip the color ramp. Would it be possible to perhaps define the data type during import to accommodate various data types or perhaps select the flipped color ramp when the emission mode is switched off? It could make it easier for inexperienced EM users to use the plugin.

- It was not completely clear to me whether it is possible to render a single/multiple EM slices using the inverted (TEM-like) contrast. For example, XY, XZ, YZ ortho slices across the volume. The manuscript contains: "This visualization is also supported in Blender, allowing for arbitrary selections of viewing angles (Fig 2B).", but it is not clear how to achieve that.

- In 3D microscopy, it is quite common to have data with anisotropic voxels. As a result, the surfaces may require smoothing. I was not able to quickly find a way to smooth the surfaces (at least smooth modifiers for surfaces did not work for me). Is it possible to apply smoothing during the import of labels, or alternatively, smoothing of the generated surfaces can be a topic for an additional YouTube video.

- It is also typical to have somewhat custom color maps for materials. It would be great if the plugin remembers the previously used color map for labels.

- The pixel size edit box rounds up the values to 2 digits after the dot. Could it be changed to accommodate 3 or 4 digits as the units are μm .

- Import is not working when:

- Start Blender

- Select Data storage: with project

- Overwrite files: on, set env: on, chunked: on

- Select a file to import

- Save Blender file

- Pressing the Load button gives an error: "Empty data directory - please save the project first before using With Project saving."

- I was not able to play the downloaded supplementary video 3 using my VLC media player, while it was working fine in a browser. The video can be opened but looks distorted and heavily zoomed in. It may need to be re-saved from a video editor.
- References 12 and 16 are URL links instead of proper references to articles.

2. Significance:

Significance (Required)

This work effectively bridges a gap in the availability of tools for 3D microscopy dataset visualization. While many visualization programs exist, the high-quality ones are often expensive and thus not accessible to all researchers. The integration of Blender with Microscopy Nodes democratizes access to high-quality 3D visualization, enabling researchers to explore datasets and models from multiple perspectives, potentially leading to new discoveries and enhancing the understanding of key study findings. Despite its limitations, my experience with the plugin was engaging and useful. I would like to thank the authors for such useful work!

Limitations:

- There remains a steep learning curve associated with using Microscopy Nodes, primarily due to Blender's complexity. More comprehensive tutorials could help mitigate this.
- The conversion of imported images to Blender's internal 32-bit format results in a 4x increase in data size for 8-bit datasets.
- Managing moderate-sized volumes (5-10 GB) can be challenging without clear strategies for effective handling.
- The import of Zarr-datasets over the net is notably slow.

Audience: The plugin is suitable for a broad audience with a basic understanding of 3D visualization concepts, providing a solid foundation for exploring Blender's extensive features and options for optimal visualizations.

Reviewer expertise: Light microscopy, electron microscopy, image segmentation and analysis, software development, no experience with Blender

3. How much time do you estimate the authors will need to complete the suggested revisions:

Estimated time to Complete Revisions (Required)

(Decision Recommendation)

Between 1 and 3 months

4. Review Commons values the work of reviewers and encourages them to get credit for their work. Select 'Yes' below to register your reviewing activity at Web of Science Reviewer Recognition Service (formerly Publons); note that the content of your review will not be visible on Web of Science.

Yes

Review #2

1. Evidence, reproducibility and clarity:

Evidence, reproducibility and clarity (Required)

****Summary:****

The article introduces Microscopy Nodes, a Blender add-on designed to simplify the loading and visualization of 3D microscopy data. It supports TIF and OME-Zarr images, handling datasets with up to five dimensions. The authors present different visualization modes, including volumetric rendering, isosurfaces, and label masks, demonstrating the application in light and electron microscopy. They provide examples using expansion microscopy, electron microscopy, and real-time imaging, highlighting how the tool enhances scientific communication and interactive visualization.

****Comments:****

However, some key aspects could be improved to enhance usability and reproducibility:

Example datasets: The images used in the YouTube tutorials were not accessible, making it difficult to reproduce the workflows shown in the figures and tutorials. It would be helpful if the authors provided direct links to the datasets or ensured that the same examples used in the tutorials were readily available for replication.

Input file specifications: The article does not clearly detail how input files should be formatted. Many users will pre-visualize images in Fiji to convert their original images to a

compatible format. It would be beneficial to specify which formats are supported for hyperstack creation, including details on bit depth, dimension ordering, label formats, and metadata compatibility, if applicable.

Hardware requirements: The article does not discuss RAM or hardware constraints in detail. In testing, attempting to load two images into the same project caused the program to freeze (tested on Mac M1). Specifying hardware requirements and limitations would help users manage expectations when working with large datasets.

2. Significance:

Significance (Required)

General Assessment:

One of the major strengths of this work is its seamless compatibility with Blender, a powerful and widely used animation and 3D rendering tool. Integrating advanced visualization techniques from the animation and graphics industry into scientific imaging opens new possibilities for presenting complex microscopy data in an intuitive and accessible way. Additionally, the support for OME-Zarr is particularly valuable, as this format represents a major shift in bioimaging towards scalable, cloud-compatible, and standardized data storage solutions. The adoption of OME-Zarr facilitates large-scale data handling and improves interoperability across imaging platforms, making this integration a significant step forward for the field.

Overall, the greatest strength of the tool lies in its flexibility for rendering microscopy data, but its accessibility for users without Blender experience might be a challenge.

Advance in the Field

This work introduces a novel solution to the visualization challenges in microscopy by leveraging Blender's advanced rendering capabilities.

Audience

This paper will be of interest to:

Bioimage researchers seeking to enhance their microscopy data visualization.

Image analysis tool developers interested in integrating advanced visualization into their workflows.

Field of Expertise

This review is based on expertise in image analysis, segmentation, and 3D biological data visualization.

3. How much time do you estimate the authors will need to complete the suggested revisions:

Estimated time to Complete Revisions (Required)

(Decision Recommendation)

Less than 1 month

4. Review Commons values the work of reviewers and encourages them to get credit for their work. Select 'Yes' below to register your reviewing activity at Web of Science Reviewer Recognition Service (formerly Publons); note that the content of your review will not be visible on Web of Science.

Yes

Review #3

1. Evidence, reproducibility and clarity:

Evidence, reproducibility and clarity (Required)

The paper "Microscopy Nodes: Versatile 3D Microscopy Visualization with Blender" presents an easy and accessible approach for microscopists and microscopy users to visualize their data in a different and more controlled way. The authors have developed a plug-in script that enables the integration of complex 3D datasets into Blender, a widely used software for 3D visualization and illustration. By leveraging Blender's advanced rendering engine, the plug-in provides greater control over the scene, environment and presentation of the 3D data.

I believe that this development, especially when combined with additional analysis tools can be of a great value for microscopist and advanced users to presenting their 3D data sets.

However, at this stage, the paper does not seem to fully demonstrate the benefits of using Microscopy Nodes. To enhance the paper impact, it would be helpful for the authors to further emphasize and provide examples of how Blender's rendering specifically improves data presentation and, in turn, enhances the understanding of the data compared to existing solutions.

Specifically, the authors claim at the end of the introduction that their development provides powerful tools for high-quality, visually compelling presentations, enabling "more effective communication of 3D biological data." I believe this statement should be supported by a figure comparing currently available visualization methods and demonstrating how using Blender enhances data presentation and by which enhances the communication of the results.

Additionally, at the end of the first paragraph of the results, the authors say:

"These options allow us to combine the data and its analyzed interpretation in the same representation with Microscopy Nodes."

However, this capability already exists in currently available software. Aside from now being able to achieve this in Blender, what additional benefits does it offer?

In the last sentence of the second paragraph of the results, it is stated: "Blender powered by Microscopy Nodes: the ability to combine microscopy data with any 3D illustration in the same 3D environment."

Could you please elaborate on the accuracy of the models that can be built and provide guidelines for achieving this using the data coordinates imported by Microscopy Nodes? If the illustrations are purely freehand and do not require specific accuracy, it would be helpful to clarify the advantages of creating them within the same environment rather than separately, as many scientists currently do.

Additionally, if the inclusion of 3D model illustrations is one of the key advantages of using Blender, I believe it would be beneficial to present this in a figure rather than only in the supplementary video.

2. Significance:

Significance (Required)

The significance of the paper at this stage is primarily technical and mainly relevant to the field of microscopy

My field of expertise is microscopy and 3D visualization of models using mainly Maya3D and AMIRA.

3. How much time do you estimate the authors will need to complete the suggested revisions:

Estimated time to Complete Revisions (Required)

(Decision Recommendation)

Less than 1 month

Yes

Response to the Reviews

We thank the reviewers for their input and detailed feedback, which has helped us improve both the manuscript and the Microscopy Nodes software. Based on the comments, we have implemented new features, currently available as version 2.2.1 of Microscopy Nodes. We have edited the text and figures of the manuscript to reflect these changes and add clarification where needed. To highlight how we have implemented this, we here show our responses in pink and indented, the original comments in black.

Reviewer #1

Evidence, reproducibility and clarity

The work by Gros et al. presents a paper introducing Microscopy Nodes, a new plugin for Blender 3D visualization software designed to import and visualize multi-dimensional (up to 5D) light and electron microscopy datasets. Given that Blender is not directly suited for such tasks, this plugin significantly simplifies the process, making its visualization engine accessible to a wide range of researchers without prior knowledge of Blender. The plugin supports importing volumes and labels from generic TIF or modern OME-Zarr image formats and includes supplementary video tutorials on YouTube to facilitate basic understanding of the visualization workflows.

Major comments:

- The manuscript suggests that Microscopy Nodes can easily handle large datasets, as evidenced by the showcases. However, in my personal tests, I was unable to import a moderate TIF stack of about 5GB, which is considerably smaller than the showcased datasets. Post-import, a data cube was displayed, but the Blender interface became unresponsive. The manuscript should include a section stating limitations and addressing issues and providing suggestions for visualization of large datasets.

We want to thank the reviewer for this valuable comment, which led us to find a core issue in Blender's large data handling. Specifically, Blender's rasterized pipeline causes issues with > 4 GiB of data loaded. This issue does not occur in the raytraced (Cycles) renderer, which is why we had not previously encountered it.

To address this, we have extended the reloading workflow of Microscopy Nodes to provide a workaround for this. If the data is larger than 4 Gibibytes (GiB) (per timepoint, or per timepoint per channel), Microscopy Nodes now automatically downsamples these data during import. While using these downsampled options is recommended for adjusting the visualization settings, the user can then still make their animation and reload their data to the largest scale for the final render by using the raytraced (Cycles) renderer. Additionally, we have raised this bug with the core Blender developers, and hope to work this out in the long term (blender/blender#136263).

We reflect these changes in the manuscript in the segment:

“Blender currently has a notable limitation that its default ‘quick’ rasterized rendering engines (such as ‘EVEE’, but also the viewport ‘Surface’ and ‘Wireframe’ modes) do not support more than 4 Gibibytes (GiB) of volumetric data. The raytracing render mode ‘Cycles’, however, can handle large volumetric data. To allow users with large data to flexibly use Microscopy Nodes, we implemented a reloading scheme, where one first loads a smaller version of the data (under 4 GiB per timeframe for all loaded channels combined) - and only upon final render in Cycles, exchange it for the full/larger scale copy (Fig 3A). This downscaling of data offers additional benefits as it allows for fast adjustment of the render settings on e.g. a personal computer which can eventually be transferred to a larger workstation or HPC cluster for the final render at full resolution. This feature is critical as working in Cycles with larger files requires sufficient RAM to fit the (temporary) VDB files comfortably. For example, multiple figures in this manuscript were made on a 32GB RAM M1 Macbook Pro (Fig 1A, Video SV1, Fig 1D, Figure 2A-D, Fig S2A-B), but for larger data or long movies the movies were made on workstations or prepared on a laptop and then transferred to an HPC cluster for final rendering.”

- The feature of importing Zarr-datasets over HTTP is great, but the import process was very slow in my tests, even on a robust network. For reference, loading 1.8 GB of the PRPE1_4x dataset at s1 level took 52 minutes. This raises concerns about potential code issues and general usability of the suggested workflow.

We believe that this loading time may have been caused by the same issue that plagued all of our datasets of >4GB outside of the raytraced mode, as we have not seen loading issues like that. Moreover, Microscopy Nodes now supports Zarr version to Zarr 3/OME-Zarr 0.5, which allows ‘sharded’ Zarr datasets, which should be even faster at loading large blocks of data at the same time, as Microscopy Nodes does.

- The onsite documentation is a bit outdated and fails to fully describe the plugin settings.

We have updated our documentation to offer new written tutorials, which include full start-up tutorials, but also for some key extra instructions.

- The YouTube tutorials feature an outdated version of the plugin, which could confuse the general microscopy audience. These should be updated to better align with the current plugin functionality. Additionally, using smaller, easily accessible datasets for these tutorials would improve user testing experiences. Hosting complete (downsampled) demo project folder on platforms like zenodo.org could also enhance usability of such tutorials.

We have made a new series of YouTube tutorials that align with the current interface of Microscopy Nodes. These tutorials include public datasets, allowing users to follow along

easily. We have chosen to also retain the older tutorials for users running legacy versions of the plugin, as they cover different workflows.

- The manuscript describes a novel dataset used in Fig. 2, but no reference is provided. Additionally, practical implementation of the coloring description for Fig. 2D can be unclear for inexperienced users, necessitating either step-by-step instructions or the provision of downsampled Blender files to aid understanding.

We have now shared the OME-Zarr address in the text (https://uk1s3.embassy.ebi.ac.uk/idr/share/microscopynodes/FIBSEM_dino_masks.zarr), and included this both in the manuscript and the tutorials. Additionally, to guide the implementation and explain the logic behind the coloring we introduced additional panels in Fig S1 and Fig S2 to showcase the shader setups used for this image.

[OPTIONAL] When importing labels, they can be assigned to individual materials only if initially split into multiple color channels. It would be great if the same logic is implemented when those materials are provided as indices within a single color channel. There can be a switch to define the logic used during the import process: e.g. the current one, when the objects are just colored based on a color map, or when they are arranged as individual materials as done when labels are imported from multiple color channels.

We agree with the reviewer and to address this concern with the update to version 2.2, we have implemented a new colorpicking system (See Fig 3B, inset 3, Fig 3C), this allows users to choose between a single color, various continuous, or categorical color maps.

Minor comments:

- The manuscript shows nice visualizations of time series, light, and electron microscopy datasets, but in its current state, it is targeted more for light microscopy, where the signal is white. On the other hand, many EM datasets are rendered in inverted contrast (TEM-like), where the signal is black. To render such volume properly, it is needed to go into the Shading tab and flip the color ramp. Would it be possible to perhaps define the data type during import to accommodate various data types or perhaps select the flipped color ramp when the emission mode is switched off? It could make it easier for inexperienced EM users to use the plugin.

To address this, we include new default settings, with 'invert colormaps on load' option in the preferences, and default colors per channel (See Fig S4). We have also implemented a new color picking system in version 2.2 (See Fig 3B, inset 3, Fig 3C) that hopefully makes it easier before and after load to change colors.

- It was not completely clear to me whether it is possible to render a single/multiple EM slices using the

inverted (TEM-like) contrast. For example, XY, XZ, YZ ortho slices across the volume. The manuscript contains: "This visualization is also supported in Blender, allowing for arbitrary selections of viewing angles (Fig 2B).", but it is not clear how to achieve that.

We introduced an additional explanation in Fig S1A and added a separate density window in the default shader to make this opaque view easier. To get a single slicing plane, users can reduce the scale of the slicing cube in one axis, at it is now also explained in Fig S2B.

- In 3D microscopy, it is quite common to have data with anisotropic voxels. As a result, the surfaces may require smoothing. I was not able to quickly find a way to smooth the surfaces (at least smooth modifiers for surfaces did not work for me). Is it possible to apply smoothing during the import of labels, or alternatively, smoothing of the generated surfaces can be a topic for an additional YouTube video.

The smoothness of the loaded masks can be indirectly affected in the preferences by changing the mesh resolution (changing the relative amount of vertices per pixel), but can be further affected by operations such as the Blender "Smooth" or e.g. the "Smooth by Laplacian" modifiers. To guide the users in doing so, we have included instructions for smoothing in the written tutorials on the website

https://aafkegros.github.io/MicroscopyNodes/tutorials/surface_smoothing/ .

- It is also typical to have somewhat custom color maps for materials. It would be great if the plugin remembers the previously used color map for labels.

We have implemented new Preference settings, which include default colors and colormaps per channel, improving customization and reproducibility. This new option is described in Figure S4.

- The pixel size edit box rounds up the values to 2 digits after the dot. Could it be changed to accommodate 3 or 4 digits as the units are um.

Blender's interface truncates the display, but stores higher-precision values internally, and become visible when users click or edit the values. We have added support for alternative pixel units to reduce the impact of the truncation.

- Import is not working when:
 - Start Blender
 - Select Data storage: with project
 - Overwrite files: on, set env: on, chunked: on
 - Select a file to import
 - Save Blender file

- Pressing the Load button gives an error: "Empty data directory - please save the project first before using With Project saving."

We thank the reviewer for finding this bug which is now fixed in version 2.2.

- I was not able to play the downloaded supplementary video 3 using my VLC media player, while it was working fine in a browser. The video can be opened but looks distorted and heavily zoomed in. It may need to be re-saved from a video editor.

We have recompiled this video.

- References 12 and 16 are URL links instead of proper references to articles.

Thanks for catching this mistake in our bibliography. We have corrected this.

Significance

This work effectively bridges a gap in the availability of tools for 3D microscopy dataset visualization. While many visualization programs exist, the high-quality ones are often expensive and thus not accessible to all researchers. The integration of Blender with Microscopy Nodes democratizes access to high-quality 3D visualization, enabling researchers to explore datasets and models from multiple perspectives, potentially leading to new discoveries and enhancing the understanding of key study findings. Despite its limitations, my experience with the plugin was engaging and useful. I would like to thank the authors for such useful work!

Limitations:

- There remains a steep learning curve associated with using Microscopy Nodes, primarily due to Blender's complexity. More comprehensive tutorials could help mitigate this.
- The conversion of imported images to Blender's internal 32-bit format results in a 4x increase in data size for 8-bit datasets.
- Managing moderate-sized volumes (5-10 GB) can be challenging without clear strategies for effective handling.
- The import of Zarr-datasets over the net is notably slow.

Audience: The plugin is suitable for a broad audience with a basic understanding of 3D visualization concepts, providing a solid foundation for exploring Blender's extensive features and options for optimal visualizations.

Reviewer expertise: Light microscopy, electron microscopy, image segmentation and analysis, software development, no experience with Blender

Reviewer #2

Evidence, reproducibility and clarity

Summary:

The article introduces Microscopy Nodes, a Blender add-on designed to simplify the loading and visualization of 3D microscopy data. It supports TIF and OME-Zarr images, handling datasets with up to five dimensions. The authors present different visualization modes, including volumetric rendering, isosurfaces, and label masks, demonstrating the application in light and electron microscopy. They provide examples using expansion microscopy, electron microscopy, and real-time imaging, highlighting how the tool enhances scientific communication and interactive visualization.

Comments:

However, some key aspects could be improved to enhance usability and reproducibility:

Example datasets: The images used in the YouTube tutorials were not accessible, making it difficult to reproduce the workflows shown in the figures and tutorials. It would be helpful if the authors provided direct links to the datasets or ensured that the same examples used in the tutorials were readily available for replication.

We created new and updated tutorials and for all new tutorials, the data is now easily available from an S3 server.

Input file specifications: The article does not clearly detail how input files should be formatted. Many users will pre-visualize images in Fiji to convert their original images to a compatible format. It would be beneficial to specify which formats are supported for hyperstack creation, including details on bit depth, dimension ordering, label formats, and metadata compatibility, if applicable.

We have added new documentation on this on the website and in the manuscript. The addon can take 8, 16, and 32 bit data, and any dimension order (with the letters tzcxy) and pixel size. Dimension order and pixel size can be edited in the GUI. This is reflected in the manuscript in the rewritten section in Design and Implementation:

“It can handle 8bit to 32bit integer and floating point data, although all data types will be resaved into 32bit floating point VDB files, which can cause temporary files to take up more space than the original. Microscopy Nodes loads 2D to 5D files of containing data across time, z, y, x and channels, in arbitrary order (can be remapped in the user interface as well, Fig 3B, inset 2). To focus on relevant data, users can clip the time axis, which can be useful for long videos.”

Hardware requirements: The article does not discuss RAM or hardware constraints in detail. In testing, attempting to load two images into the same project caused the program to freeze (tested on Mac M1). Specifying hardware requirements and limitations would help users manage expectations when working with large datasets.

We have since found a limitation in the Blender engine that indeed limits the amount of data loaded (see also comment by Reviewer 1). Currently, rasterized engines are capped at 4 GiB, and only the raytraced engine can handle larger data. As such, the Microscopy Nodes pipeline, where one works with small images until it is time to render a final version, and the data is only exchanged for the final render, is still viable. To make this easier, we now also included optional downscaling for Tif images. This is described in the rewritten section on Design and Implementation:

“Blender currently has a notable limitation that its default ‘quick’ rasterized rendering engines (such as ‘EVEE’, but also the viewport ‘Surface’ and ‘Wireframe’ modes) do not support more than 4 Gibibytes (GiB) of volumetric data. The raytracing render mode ‘Cycles’, however, can handle large volumetric data. To allow users with large data to flexibly use Microscopy Nodes, we implemented a reloading scheme, where one first loads a smaller version of the data (under 4 GiB per timeframe for all loaded channels combined) - and only upon final render in Cycles, exchange it for the full/larger scale copy (Fig 3A). This downscaling of data offers additional benefits as it allows for fast adjustment of the render settings on e.g. a personal computer which can eventually be transferred to a larger workstation or HPC cluster for the final render at full resolution. This feature is critical as working in Cycles with larger files requires sufficient RAM to fit the (temporary) VDB files comfortably. For example, multiple figures in this manuscript were made on a 32GB RAM M1 Macbook Pro (Fig 1A, Video SV1, Fig 1D, Figure 2A-D, Fig S2A-B), but for larger data or long movies the movies were made on workstations or prepared on a laptop and then transferred to an HPC cluster for final rendering.”

Significance

General Assessment:

One of the major strengths of this work is its seamless compatibility with Blender, a powerful and widely used animation and 3D rendering tool. Integrating advanced visualization techniques from the animation and graphics industry into scientific imaging opens new possibilities for presenting complex microscopy data in an intuitive and accessible way. Additionally, the support for OME-Zarr is particularly valuable, as this format represents a major shift in bioimaging towards scalable, cloud-compatible, and standardized data storage solutions. The adoption of OME-Zarr facilitates large-scale data handling and improves interoperability across imaging platforms, making this integration a significant step forward for the field.

Overall, the greatest strength of the tool lies in its flexibility for rendering microscopy data, but its accessibility for users without Blender experience might be a challenge.

Advance in the Field

This work introduces a novel solution to the visualization challenges in microscopy by leveraging Blender's advanced rendering capabilities.

Audience

This paper will be of interest to:

Bioimage researchers seeking to enhance their microscopy data visualization.

Image analysis tool developers interested in integrating advanced visualization into their workflows.

Field of Expertise

This review is based on expertise in image analysis, segmentation, and 3D biological data visualization.

Reviewer #3 (Evidence, reproducibility and clarity (Required)):

The paper "Microscopy Nodes: Versatile 3D Microscopy Visualization with Blender" presents an easy and accessible approach for microscopists and microscopy users to visualize their data in a different and more controlled way. The authors have developed a plug-in script that enables the integration of complex 3D datasets into Blender, a widely used software for 3D visualization and illustration. By leveraging Blender's advanced rendering engine, the plug-in provides greater control over the scene, environment and presentation of the 3D data.

I believe that this development, especially when combined with additional analysis tools can be of a great value for microscopist and advanced users to presenting their 3D data sets.

However, at this stage, the paper does not seem to fully demonstrate the benefits of using Microscopy Nodes. To enhance the paper impact, it would be helpful for the authors to further emphasize and provide examples of how Blender's rendering specifically improves data presentation and, in turn, enhances the understanding of the data compared to existing solutions.

Specifically, the authors claim at the end of the introduction that their development provides powerful tools for high-quality, visually compelling presentations, enabling "more effective communication of 3D biological data." I believe this statement should be supported by a figure comparing currently available visualization methods and demonstrating how using Blender enhances data presentation and by which enhances the communication of the results.

Additionally, at the end of the first paragraph of the results, the authors say:

"These options allow us to combine the data and its analyzed interpretation in the same representation with Microscopy Nodes."

However, this capability already exists in currently available software. Aside from now being able to achieve this in Blender, what additional benefits does it offer?

We now include a new Table 1, to showcases which requirements for visualizing complex biological data are available in different visualization software, and discuss this in the text:

"Although several tools for 3D visualization of bioimages already exist and offer essential features for microscopy data (Table 1), many are proprietary, and open-source alternatives often struggle to deliver a comprehensive user experience, such as advanced animation and annotation controls. Proprietary solutions may offer some of these capabilities, but they are

frequently limited by licensing costs, platform restrictions, and a lack of customizability. In contrast, Blender is a mature, well-supported open-source platform with a large community of developers that excels in both animation and visualization. By integrating microscopy-specific functionality through Microscopy Nodes, Blender becomes a uniquely powerful solution that bridges the gap between high-end graphics capabilities and the specialized needs of bioimage visualization.”

Additionally, we attempted to remake Figure 2C and 2D in the EM-field standard software Amira:

Where A is an attempt to recreate Figure 2C, and B an attempt to recreate Figure 2D. Here we can see that without light scattering, it is very hard to see the depth in the nucleus, and the semi-transparent masks do show each other behind them, but cannot interact with the volume.

We chose not to include this in the actual manuscript, as we are not experts at the Amira software, and will, by the nature of this manuscript, present a challenge that Blender is especially good at, such as here the combination of scattering light and semitransparent masks.

In the last sentence of the second paragraph of the results, it is stated: "Blender powered by Microscopy Nodes: the ability to combine microscopy data with any 3D illustration in the same 3D environment."

Could you please elaborate on the accuracy of the models that can be built and provide guidelines for achieving this using the data coordinates imported by Microscopy Nodes? If the illustrations are purely freehand and do not require specific accuracy, it would be helpful to clarify the advantages of creating them within the same environment rather than separately, as many scientists currently do.

Additionally, if the inclusion of 3D model illustrations is one of the key advantages of using Blender, I believe it would be beneficial to present this in a figure rather than only in the supplementary video.

We thank the reviewer for this comment and agree that in the previously submitted version of Microscopy Nodes, it was very difficult to align objects accurately, as the coordinate space was not transparent. A hurdle in this was the fact that Blender only works well with the unit 'meters'. To address this issue, we now provide a choice of mapping the physical size to meters, as shown in the new interface (See Fig 3B, inset 5). Here the user can choose from the default 'px -> cm' (this will always look fine for a quick look) to options such as 'nm -> m' or ' μm -> m', which, combined with the new choice for adjusting the object origin upon load, allow users to treat the Blender coordinate space as based on the actual physical scales. Additionally, other Blender addons, such as Molecular Nodes (Reference 25 of the manuscript), also allow for accurate localization for cryo-EM datasets.

We appreciate the note that we should more clearly display the ability to show our illustrations and the data together in the figure and have added a visualization to show this in Figure 1C.

Reviewer #3 (Significance (Required)):

The significance of the paper at this stage is primarily technical and mainly relevant to the field of microscopy

My field of expertise is microscopy and 3D visualization of models using mainly Maya3D and AMIRA.

Dear Dr. Banterle,

Thank you for the submission of your revised manuscript. We have now received the enclosed reports from the referees and I am happy to say that all support its publication now. Referee 1 still has a few more minor suggestions that I would like you to incorporate before we can proceed with the official acceptance of your manuscript.

A few editorial requests will also need to be addressed:

- Your ms has 3 main figures and will therefore be published as a short report. Please combine the results and discussion section to comply with our short reports format.
- Please upload the ms as a Word file with figure legends, while the figures should be uploaded as individual high quality production Figure files.
- Please add up to 5 keywords to the ms file.
- Please rename the "Code Availability section" to "Data Availability Section" and place it before the Acknowledgments.
- Please add a "Disclosure and Competing Interest Statement" to the ms file.
- The REFERENCE format needs to be alphabetical, not numerical; et al needs to be used after 10 author names; DOIs should only be used for preprints and datasets that have not been published yet. Please use the EMBO reports reference style.
- Please send us with your final ms a complete author checklist, which you can download from our author guidelines <<https://www.embopress.org/page/journal/14693178/authorguide>>. The completed author checklist will also be part of your transparent peer-review file.
- The FUNDING INFO the European Molecular Biology Laboratory also needs to be entered in our online ms submission system as a separate funder.
- The APPENDIX FILE nomenclature is not correct - "supplemental" should not be used; it needs to be called Appendix and the figures and their callouts should be Appendix Figure S1, etc.; the movie legends need to be removed from the file; a title page with a Table of Content with page numbers are also needed; "Supplemental Note 1" is also not a correct title, please correct.
- A figure callout is missing for Figure S1/Appendix Figure S1.
- The MOVIES nomenclature needs to be corrected to Movie EV1-EV4 in all places: source file names, titles in the system, legends and callouts in the ms; the legends should be removed from the suppl. file and each should be provided in a separate file (Word or readme.txt) and then zipped up with its corresponding movie so that we have 4 zip folders upldd (Movie EV1-EV4).
- The Methods section should include a separate Reagents and Tools Table file (listing key reagents, experimental models, software and relevant equipment and including their sources and relevant identifiers) followed by a Methods and Protocols section in which authors should describe their methods using a step-by-step protocol format with bullet points, to facilitate the adoption of the methodologies across labs. More information on how to adhere to this format as well as downloadable templates (.docx) for the Reagents and Tools Table can be found in our author guidelines: <<https://www.embopress.org/page/journal/14693178/authorguide#manuscriptpreparation>>.
- The manuscript sections should be in the following order: Title page - Abstract & Keywords - Introduction - Results - Discussion - Methods - Data Availability - Acknowledgments - Disclosure Statement & Competing Interests - References - Figure Legends - (Main Tables with legends if applicable) - Expanded View Figure Legends.

EMBO press papers are accompanied online by A) a short (1-2 sentences) summary of the findings and their significance, B) 2-3 bullet points highlighting key results and C) a synopsis image that is exactly 550 pixels wide and 200-600 pixels high (the height is variable). The synopsis image should provide a sketch of the major findings, like a graphical abstract. Please note that text needs to be readable at the final size. Please send us this information along with the final manuscript.

Referee #1:

The authors have satisfactorily addressed all the concerns I raised in my previous review. They have made significant efforts to optimize the software, as well as to update various learning materials and documentation pages. I recommend publication of this work after the authors address the minor revision points listed below.

Minor points

1. Although the authors re-rendered Movie 3, it still does not play correctly for me. It appears that the video was rendered at a resolution of 1920x1920 pixels, which is non-standard. As a result, the video looks distorted and is not convincing. I suspect there was a mistake in the output settings: instead of rendering at the standard Full HD resolution of 1920x1080 pixels, the height field may have been incorrectly set to 1920.
2. The same issue applies to Video 4, which was also rendered at 1920x1920 pixels.
3. Since I requested these changes, I would also suggest that the Blender screenshot shown in Supplementary Figure 2A be stretched to the full-page width. Otherwise, the text and details are not easily visible.

Referee #2:

Thank you for the opportunity to review the revised version of this manuscript. I have carefully read the authors' response and the updated manuscript. I am pleased to see that the authors have thoroughly addressed all of my comments and implemented the suggested improvements, including clearer dataset availability, detailed input file specifications, and comprehensive documentation on hardware requirements and limitations.

I have no further concerns, and I believe the manuscript is now suitable for publication.

Referee #3:

The authors have addressed all my comments, and from my perspective the paper is ready for publication without further revisions

Response to referees.

We thank the reviewers for the careful reading and helpful comments that have improved the manuscript.

Referee #1: The authors have satisfactorily addressed all the concerns I raised in my previous review. They have made significant efforts to optimize the software, as well as to update various learning materials and documentation pages. I recommend publication of this work after the authors address the minor revision points listed below.

*Minor points*1. Although the authors re-rendered Movie 3, it still does not play correctly for me. It appears that the video was rendered at a resolution of 1920x1920 pixels, which is non-standard. As a result, the video looks distorted and is not convincing. I suspect there was a mistake in the output settings: instead of rendering at the standard Full HD resolution of 1920x1080 pixels, the height field may have been incorrectly set to 1920.

2. The same issue applies to Video 4, which was also rendered at 1920x1920 pixels.

While the 1920x1920 pixel render size is non-standard, the videos were indeed designed to be in square aspect ratio, and we have not been able to reproduce the issues with playing. The EMBO Reports editors said it would be appropriate to keep the files as originally submitted.³

Since I requested these changes, I would also suggest that the Blender screenshot shown in Supplementary Figure 2A be stretched to the full-page width. Otherwise, the text and details are not easily visible.

We agree that this would improve legibility and have changed the figure accordingly.

Referee #2: Thank you for the opportunity to review the revised version of this manuscript. I have carefully read the authors' response and the updated manuscript. I am pleased to see that the authors have thoroughly addressed all of my comments and implemented the suggested improvements, including clearer dataset availability, detailed input file specifications, and comprehensive documentation on hardware requirements and limitations. I have no further concerns, and I believe the manuscript is now suitable for publication.

Referee #3: The authors have addressed all my comments, and from my perspective the paper is ready for publication without further revisions

Dr. Niccolo Banterle
European Molecular Biology Laboratory (EMBL)
Cell Biology and Biophysics Unit
Meyerhofstr. 1
Heidelberg 69117
Germany

Dear Niccolo,

I am very pleased to accept your manuscript for publication in the next available issue of EMBO reports. Thank you for your contribution to our journal.
